# Mechanical and Thermal Properties of the High Thermal Conductivity Steel (HTCS) Additively Manufactured via Powder-Fed Direct Energy Deposition

**DOI:** 10.3390/mi14040872

**Published:** 2023-04-18

**Authors:** Jong-Youn Son, Ki-Yong Lee, Gwang-Yong Shin, Chang-Hwan Choi, Do-Sik Shim

**Affiliations:** 1Department of Mechanical Engineering, Stevens Institute of Technology, 1 Castle Point Terrace, Hoboken, NJ 07030, USA; json4@stevens.edu; 2Automotive Materials & Components R&D Group, Korea Institute of Industrial Technology, 9, Cheomdanventure-ro 108beon-gil, Buk-gu, Gwangju 61007, Republic of Korea; kylee@kitech.re.kr (K.-Y.L.); shin2007@kitech.re.kr (G.-Y.S.); 3Division of Mechanical Engineering, Korea Maritime and Ocean University, 727, Taejong-ro, Yeongdo-gu, Busan 49112, Republic of Korea

**Keywords:** direct energy deposition, high thermal conductivity steel, response surface methodology, thermal conductivity, mechanical property, elevated temperature

## Abstract

High thermal conductivity steel (HTCS-150) is deposited onto non-heat-treated AISI H13 (N-H13) via powder-fed direct energy deposition (DED) based on the response surface methodology (RSM) to enhance the mechanical properties and thermal conductivity of N-H13, which is generally used as a hot-work tool steel. The main process parameters of the powder-fed DED are priorly optimized to minimize defects in the deposited regions and, therefore, to obtain homogeneous material properties. The deposited HTCS-150 is comprehensively evaluated through hardness, tensile, and wear tests at the different temperatures of 25, 200, 400, 600, and 800 °C. Compared to conventionally heat-treated (quenched and tempered) H13 (HT-H13), the hardness of the additively manufactured HTCS-150 slightly increases at 25 °C, whereas it does not show any significant difference above 200 °C. However, the HTCS-150 deposited on N-H13 shows a lower ultimate tensile strength and elongation than HT-H13 at all tested temperatures, and the deposition of the HTCS-150 on N-H13 enhances the ultimate tensile strength of N-H13. While the HTCS-150 does not show a significant difference in the wear rate below 400 °C compared to HT-H13, it shows a lower wear rate above 600 °C. The HTCS-150 reveals a higher thermal conductivity than the HT-H13 below 600 °C, whereas the behavior is reversed at 800 °C. The results suggest that the HTCS-150 additively manufactured via powder-fed direct energy deposition can enhance the mechanical and thermal properties of N-H13, including hardness, tensile strength, wear resistance, and thermal conductivity in a wide range of temperatures, often superior to those of HT-H13.

## 1. Introduction

Hot-work tool steels are widely used in hot working processes such as forging, die casting, and hot stamping processes. They are exposed to severe working conditions with the rapid changes in temperature. Hot-work tool steels are repeatedly heated by heated or molten materials and cooled by lubricants and coolants. The repetitive heating and cooling cycles consequently generate thermal stresses inside the part and fatigue failures on the surface of the part. Partial wears or defects on the tool steels can concentrate the thermal stress and accelerate fatigue failures so that wear resistance and surface hardness are considered as crucial properties of the hot working tool steel. Moreover, thermal conductivity is another critical property of the hot-work tool steel to reduce the lead time of production processes and production costs in manufacturing. Thus, the industry has become interested in simultaneously improving the mechanical properties and thermal conductivity of the tool steel.

There have been many studies on increasing mechanical properties of the tool steels using heat treatments. However, the heat treatments have limitations on the tool steel utilized in hot working conditions because the hardened structures by the heat treatments can be annealed at hot working conditions. In addition, the hardening by heat treatments generally decreases the thermal conductivity. Thus, the industry has been exploring developing alloys with enhanced mechanical properties and thermal conductivity at high temperatures. There have been alloys reported to have both enhanced mechanical properties and thermal conductivity [1,2], and the alloys are manufactured through the powder press method with carbide forming elements in order to adjust the density of states [3], Fermi energy levels, phonon spectra, and scattering in crystalline structure [4,5]. However, the alloys are too expensive to apply to general parts compared to normal tool steels. They are also difficult to be machined at room temperature because of their excessively high hardness and strength, so the applications of such alloys have been quite limited. To address such limitations, there have recently been attempts to fabricate multiple materials and modify surface properties via additive manufacturing. Bariman et al. [6] studied the laser molten surfaces of the high thermal conductivity steel for surface modification. Dinda et al. [7] investigated laser-aided direct metal depositions of Inconel alloys regarding microstructure and thermal stability. Shim et al. [8] proposed design methods for the layer thickness setting to improve the geometric accuracy of the final part but also investigated the deposition methods of AISI M4 using a preheated substate and the mechanical properties of the deposited M4. Ingrassia et al. [9] studied the influence of parameters of the additive manufacturing process on the shape and dimensional accuracy of deposition. Hong et al. [10] investigated additively manufactured heterogeneous materials to fabricate a hot-stamping die and examine the various mechanical properties. However, most studies limitedly analyzed the principal process parameters, and there is a lack of research on analyzing the deposition characteristics by considering the process parameters. Moreover, most works for additively manufactured materials have been by evaluating the mechanical properties of deposited materials mainly at room temperature, and some of them have covered a limited range of elevated temperatures [11,12]. Thus, more extensive and systematic investigations of the mechanical and thermal properties of the additively manufactured hot-work tool steel in a wider range of temperature are necessary.

This study investigated the surface modification of hot-work tool steel via the powder-fed direct energy deposition (DED) technique to enhance the mechanical properties and thermal conductivity of the hot-work tool steel. For the surface modification, high thermal conductivity steel (HTCS-150) was partially deposited onto non-heat-treated AISI H13 (N-H13) which is generally used as hot-work tool steel. The process parameters of the powder-fed DED were optimized with the response surface methodology (RSM) to minimize defects in a deposited region. Then, the deposited HTCS-150 was comprehensively evaluated for hardness, tensile, and wear at elevated temperatures, compared to N-H13 as well as conventionally heat-treated H13 (HT-H13). Moreover, the thermal conductivity of the deposited HTCS-150 was examined at elevated temperatures to demonstrate the efficacy of the powder-fed DED process for enhancing both the mechanical properties and thermal conductivity at the same time.

## 2. Materials and Methods

### 2.1. Material and Powder-Fed DED Process

The powder of high thermal conductivity steel (HTCS-150, Rovalma S.A., Barcelona, Spain) was used for the metal powder to be deposited via the powder-fed DED process. The HTCS-150 has two times higher thermal conductivity than AISI H13 (N-H13) and the HTCS-150 powders consist of spherical particles with diameters of 50–160 μm, as shown in Figure 1. Table 1 summarizes the chemical compositions of the HTCS-150 and the N-H13.

The HTCS-150 was deposited on N-H13 substrates (100 mm × 50 mm × 15 mm) via the powder-fed DED machine (MX3, InssTek, Daejeon, Republic of Korea) using a CO_2_ laser. Figure 2 depicts the schematics of the powder-fed DED process. The DED process utilizes Ar gas as a carrier gas to inject the metal powder and as a coaxial gas to direct the metal powder to the melting pool. In the DED process, a laser beam is irradiated to a substrate to form a melting pool and the powdered material is simultaneously fed into the melting pool, as shown in Figure 2a. Thus, a single bead is deposited onto the substrate with solidification of the molten powder and the melting pool, and the single beads are continuously overlapped to the previously deposited beads. The overlapped multiple beads become a layer and the deposited layers become a structure, as shown in Figure 2b.

### 2.2. Response Surface Methodology (RSM)

The deposited structure via the DED process often shows defects such as a lack of fusion and pores being between beads or layers, as shown in Figure 3. The defects depend on the deposited geometries, which are directly affected by process parameters. The optimization of the process parameters is generally required to obtain a homogeneous quality and minimize the defects in the deposited region. Thus, in this work, the main process parameters of the powder-fed DED were analyzed using the response surface methodology (RSM). The RSM consists of a group of mathematical and statistical techniques used in the development of an adequate functional relationship between a response of interest, Y, and a number of associated control variables denoted by $X_{1},\;X_{2},\;\cdots ,\;X_{k}$ [13,14,15]. In general, multiple regression is typically used to build an empirical model for predicting the response, and the empirical model can be approximated by the following Equation (1):(1)$$Y=b_{0}+\sum_{i=k}^{k}b_{i}X_{i}+\sum_{\begin{array}{c} i,j=k\\ i\neq j \end{array}}^{k}b_{ij}X_{i}X_{j}+\sum_{i=1}^{k}b_{ii}X_{i}{}^{2}$$
where $b_{0}$ is the free term in the regression equation; $b_{1},b_{2},\cdots ,b_{k}$ are the coefficients for the linear terms; $b_{11},b_{22},\cdots ,b_{kk}$ are the coefficients for the quadratic terms; and $b_{12},b_{13},\cdots ,b_{k-1k}$ are the coefficients for the interaction terms. Table 2 lists the process parameters of the powder-fed DED. Laser power, powder feed rate, and bead spacing are generally considered as the main parameters to affect deposited geometries. Thus, these main parameters were selected as the control variables to design experimental cases, while the lack of fusion and pores in the deposition were selected as the responses. The control variables were arranged by central composite design (CCD) to design the experimental cases. Table 3 shows their bounds for the cases of the CCD. Each case deposited 5 layers with a length of 40 mm and a width of 10 mm. The lack of fusion and the pores selected as the responses were quantitatively measured on the cross-sections of each case, and their areas were calculated by the following Equation (2), respectively: (2)$$Lack\;of\;fusion\;or\;pores=\frac{Area\;of\;lack\;of\;fusion\;or\;pores\;\left({\mathrm{mm}}^{2}\right)}{Area\;of\;deposited\;region\left({\mathrm{mm}}^{2}\right)}\times 100$$

### 2.3. Hardness, Tensile, and Wear Tests at Elevated Temperatures

The deposited HTCS-150 was comprehensively evaluated for hardness, tensile, and wear at the elevated temperatures of 25, 200, 400, 600, and 800 °C. The results were compared to conventionally heat-treated H13 (HT-H13) with quenching and tempering. Each test specimen was held at the targeted temperatures for 10 min to stabilize their thermal expansion, and each test was repeated 3 times at each temperature. Figure 4 shows the schematics of the test specimens, which comprised the HTCS-150 layer deposited on the N-H13 substrate.

Hardness was measured via micro-indentation testers, including AAV-502 (Mitutoyo, Aurora, CO, USA) and QM-2 (Nikon, Tokyo, Japan). A test load of 1.9 N was applied to each indentation for 15 s, in accordance with the ASTM International Standard Test Method E384. Tensile tests were performed using the MTS 810 material test system (MTS system, Eden Prairie, MN, USA) and a heating furnace using heating coils. A tensile force was applied axially to both ends of the specimen at a rate of 2 mm/min, using stroke control, until the specimen exhibited a tensile fracture. The wear tests were performed using the Rtec MTF-5000 wear tester (Rtec Instruments, San Jose, CA, USA) with a ball-on-disc rotating type. The wear test specimen was turned in a clockwise direction at a rate of 0.157 m/s in contact with an alumina oxide ball (7.14 mm in diameter) under a normal force of 100 N, and the sliding distance was 500 m on each test. The wear losses of the specimens were measured by weighing them before and after the tests, and the wear rate was calculated by the following Equation (3):(3)$$Wear\;rate\;\left(W_{r}\right)=\frac{\Delta V}{NL}=\frac{\Delta m}{\rho NL}$$
where Δ*V*, Δm, *ρ*, *N*, and *L* are the volume loss, mass loss, density, load, and sliding distance, respectively. Those of the worn surfaces were measured using the Contour GT-X optical profilometer (Bruker, Billerica, MA, USA).

### 2.4. Thermal Conductivity at Elevated Temperatures

The thermal conductivity of the deposited HTCS-150 was calculated using Equation (4):(4)$$Thermal\;conductivity\;\left(k\right)=\alpha \cdot \rho \cdot C_{p}$$
where k is the thermal conductivity (W/m·K), α is the thermal diffusivity (${\mathrm{mm}}^{3}/s$), ρ is the density (g$/{\mathrm{cm}}^{3}$), and $C_{p}$ is the specific heat (J/kg·K). The density of specimens was measured using the Archimedes rule. The thermal diffusivity was measured using a laser flash apparatus LFA 457 (NETZSCH, Selb, Germany) at the elevated temperatures of 25, 200, 400, 600, and 800 °C, and the specimens of the thermal diffusivity were prepared to have a diameter of 12.5 mm and a thickness of 2.5 mm, as shown in Figure 5. The specimens were coated using a carbon spray to prevent the laser reflection on the surfaces of the specimens. The specific heats were determined using a differential scanning calorimetry (DSC) 404 F1 device (NETZSCH, Selb, Germany) with a pre-vacuum condition on the gas flow of 50 mL/min, and cylindrical specimens with 6 mm diameter and 1 mm thickness were prepared for measuring the specific heat. The measurements of each specimen were repeated five times.

### 2.5. Observation of Microstructures

The cross-sectional specimens of the deposited HTCS-150 were prepared using TechCut 5 (Allied High Tech Products, Inc., Compton, CA, USA) to observe the microstructures of the deposited layer. The cross-sectioned surface was polished and then etched using the nital etchant (nitric acid 5% + ethyl alcohol 95%) for several seconds. Their microstructures were observed using the HRM-300 optical microscope (Huvitz, Anyang, Republic of Korea) and the JSM-7100f (JEOL Ltd., Tokyo, Japan).

## 3. Results and Discussion

### 3.1. Regression Analysis for Minimizing Defects

To deposit multiple layers in the DED process, a height of a single layer should be predefined. Single layers of each experimental case were preferentially deposited, and their heights were measured. Equation (5) shows a single-order regression equation based on the measured data to predict a height of a single layer, depending on the process parameters. The regression analysis showed that the variance inflation factors for all variables was 1, indicating that there was no correlation between the variables. *p*-values of all the variables equal to 0, implying that the mathematical model represented by Equation (5) was significant. The coefficient of determination (R^2^), which implies how close the predicted model was to the experimental data, was 0.927 and the adjusted R^2^ was 0.914, indicating a high accuracy of the equation to predict the height of a single layer.
(5)$$Bead\;height\left(\mathrm{mm}\right)=0.0281+0.000504\;L_{p}+0.0714\;P_{fr}-0.883\;B_{space}$$
where $\;L_{p}$ is the laser power (W), $P_{fr}$ is the powder feeding rate (g/min), and $B_{space}$ is the bead spacing (mm).

After the height of a single layer was defined, multiple layers (5 layers) of each experimental case were deposited, and those of the cross-sections were observed with an optical microscope. Whereas no delamination or crack was observed in all cases, the lack of fusion and pores was observed in the deposited regions for all cases. Equations (6) and (7) show the second-order regression equations for the lack of fusion and pores in the deposited region depending on the process parameters. The regression analysis for the lack of fusion revealed a *p*-value of 0.005, an R^2^ of 0.84, and an adjusted R^2^ of 0.7. The regression analysis for the pores revealed a *p*-value of 0.008, an R^2^ of 0.83, and an adjusted R^2^ of 0.67. As the R^2^ value was close to 1, the regression equation had a highly accurate prediction. Although the adjusted R^2^ values were relatively low, they satisfied a confidence level at the *p*-values of 0.005 and 0.008, respectively. Figure 6 shows the cross-sectional views of the response surfaces based on the regression Equations (6) and (7). It derives a set of parameters that can simultaneously reduce the lack of fusion and pores by 0.185 and 0.063%, respectively. To verify the derived models, a height of a single layer was firstly predicted by the regression of Equation (5), and five layers were deposited under the derived set of parameters. The deposited single layer showed a height of 0.212 mm, and the 5 layers showed a lack of fusion of 0.297%, as shown in Figure 7. The derived models approximated the actual results relatively well, indicating the high accuracy of the predicted models.
(6)$$Lack\;of\;fusion\;\left({\mathrm{mm}}^{2}\right)\;=-5.4982-0.0091\;L_{p}+3.25\;P_{fr}+7.711\;B_{space}+L_{p}{}^{2}+0.0119\;P_{fr}{}^{2}+10.5159\;B_{space}{}^{2}\;-0.0017\;L_{p}\;P_{fr}-0.0065\;L_{p}\;B_{space}-3.1013\;P_{fr}\;B_{space}\;$$
(7)$$\;\;\;\;\;\;Pores\;\left({\mathrm{mm}}^{2}\right)=23.8115-0.0421\;L_{p}-2.3285\;P_{fr}-25.0487B_{space}+L_{p}{}^{2}-0.517\;P_{fr}{}^{2}-30.1625\;B_{space}{}^{2}\;+0.0043\;L_{p}\;P_{fr}+0.0116\;L_{p}\;B_{space}+9.415\;P_{fr}\;B_{space}$$

### 3.2. Mechanical Properties at Elevated Temperatures

#### 3.2.1. Hardness

Figure 8 shows the hardness of the HTCS-150 deposited on N-H13, compared to that on HT-H13, at elevated temperatures. At 25 °C, the deposited HTCS-150 showed a hardness of 661 HV, whereas the HT-H13 showed a lower hardness of 600 HV. As the temperature increased, the difference became less so that the hardness of the deposited HTCS-150 was around the same with that of HT-H13 at temperatures above 400 °C. The results indicated that the hardness of the HTCS-150 deposited by the powder-fed DED process should be as good as that of HT-H13 at wide ranges of temperatures. When the temperature was relatively low (below 400 °C), the HTCS-150 deposited by the powder-fed DED process could allow one to have a greater hardness than HT-H13.

#### 3.2.2. Tensile Strength and Elongation

Figure 9 shows the ultimate tensile strength (UTS) and ultimate elongation (UE) of the HTCS-150 deposited on N-H13, compared to those of HT-H13, at elevated temperatures. While the HTCS-150 deposited on N-H13 generally showed a lower UTS than HT-H13 up to 600 °C, the difference became negligible at 800 °C. Considering the relatively low UTS of the N-H13 (about 75% of that of HT-H13) used for the supporting substrate for the HTCS150 deposition, the UTS of the additively manufactured HTCS-150 via the powder-fed DED process should be notable. Meanwhile, the UE of the HTCS-150 deposited on N-H13 was lower than that of HT-H13 by 20–50% for the tested range of temperatures. This indicates that the HTCS-150 deposited on N-H13 should have a lower ductility and toughness than HT-H13, while retaining a comparable UTS.

#### 3.2.3. Wear Resistance

Figure 10 shows the wear rate of the HTCS-150 deposited on N-H13, compared to that of HT-H13, at elevated temperatures. At relatively low temperatures (up to 400 °C), the wear rate of the HTCS-150 deposited on N-H13 was not significantly different from that of HT-H13. However, the HTCS-150 deposited on N-H13 started to show the significantly lower wear rates than that of HT-H13 at 600 °C, which is the temperature where the hot-work tool steel is normally used. While the difference became more dramatic at 800 °C, it should be noted that the hardened microstructures of the deposited material can be annealed at the relatively high temperature of 800 °C. In general, wear resistance is significantly affected by the microstructure and the presence of carbides, which are determined by the difference in chemical compositions. The martensite phase in microstructure also increases the wear resistance. Whereas it is expected that both the HT-H13 and the HTCS-150 deposited on N-H13 would have stable microstructures up to 400 °C, their microstructures would start to be tempered at 600 °C and annealed at 800 °C with softening. These results suggest that the softening resistance on the HTCS-150 deposited on N-H13 should be higher than that on the HT-H13 at 600 °C and beyond. This is attributed to the chemical compositions of the HTCS-150, which includes higher molybdenum and lower vanadium than H-13. The higher molybdenum and lower vanadium content can effectively avoid the formation of large-size primary carbides and promote the formation of fine molybdenum carbides. Molybdenum retards the coarsening of nano-sized carbides and inhibits the annihilation of dislocations at elevated tempering temperatures [16,17]. Moreover, the molybdenum leads to a longer retention of the mechanical properties in steel alloys during high-temperature aging [18,19,20]. Furthermore, the additional tungsten to steel delays the dislocation recovery during tempering above 650 °C, resulting from the decreased self-diffusivity of iron [21]. The tungsten compositions lead the high-temperature wear resistance of the HTCS-150.

As shown in Figure 11a, the wear track of HT-H13 generally exhibited a rougher surface with more irregular wear scars than that of the HTCS-150 deposited on N-H13. As shown in Figure 11b, the wear track of the HTCS150 deposited on N-H13 exhibited a regular groove with little wear scar. Furthermore, as shown in Figure 11c, the wear track of the HTCS-150 deposited on N-H13 was shallower and narrower than that of the HT-H13. These results indicate that the HTCS-150 additively manufactured via the powder-fed DED process generally has a better wear resistance than HT-H13.

### 3.3. Thermal Conductivity at Elevated Temperatures

Figure 12 shows the thermal conductivities of the HTCS-150 deposited on N-H13, as-deposited HTCS-150, N-H13, and HT-H13. The thermal conductivities of the HTCS-150 deposited on N-H13 and the as-deposited HTCS-150 are around the same and do not change much with the increase in temperature. However, the thermal conductivities of N-H13 and HT-H13 increase significantly with the temperature. The thermal conductivity of N-H13 was significantly lower than that of the HTCS-150 deposited on N-H13 and the as-deposited HTCS-150 up to 200 °C, but became similar for 400–600 °C and then much greater at 800 °C. In the case of HT-H13, the thermal conductivity was even lower than that of N-H13 up to 600 °C. However, similar to that of N-H13, the thermal conductivity of HT-H13 became much greater than that of the HTCS-150 deposited on N-H13 and the as-deposited HTCS-150 at 800 °C. The differences in thermal conductivities are attributed to their microstructures. In general, heat treatments on metallic materials lead to phase transformations and increase precipitates to enhance mechanical strength. However, those structures can disrupt the electron movement to transfer a heat energy. As shown in Figure 13, the non-heat-treated N-H13 has relatively coarse precipitates in its microstructures, whereas the heat-treated HT-H13 has finer precipitates that are mostly located in grain boundaries. Consequently, the microstructural differences caused the different thermal conductivities between N-H13 and HT-H13. Meanwhile, the HTCS-150 deposited on N-H13 and the as-deposited HTCS-150 generally showed greater thermal conductivities than N-H13 and HT-H13, and it is attributed to the materials’ characteristics with molybdenum composition. Generally, molybdenum contributes to an increase in the thermal conductivity in steel alloys [3,22,23], and the HTCS-150 has larger molybdenum composition than H13. In addition, HTCS also contains carbide-forming elements of titanium, zirconium, hafnium, and niobium [2]. These elements that formed carbide networks in alloys have high melting temperatures, and interfacial carbides increase the heat transmission in the alloys [24,25]. The HTCS-150 shows a significantly higher number of carbides in its microstructure than those of N-H13 and HT-H13, as shown in Figure 14; this is mainly attributed to the higher thermal conductivity of the HTCS-150 deposited on N-H13 and as-deposited HTCS-150. However, the N-H13 and HT-H13 showed higher thermal conductivities than the HTCS-150 deposited on N-H13 and the as-deposited HTCS-150 at 800 °C. This is attributed to the phase transformation of the specimens at around 800 °C. The results suggest that HTCS-150 should not have a significant phase transformation and, hence, a change in thermal conductivity up to 800 °C.

## 4. Conclusions

In this study, the process parameters of the powder-fed DED were optimized through the response surface methodology, and the high thermal conductivity steel (HTCS-150) was deposited onto hot-work tool steel (H13). The deposited HTCS-150 was evaluated for its mechanical properties and thermal conductivity at elevated temperatures. The following conclusions can be drawn from the results.

(1)The procedures to optimize the process parameters of the power-fed DED served as an effective statistical methodology to quantitatively derive a set of process parameters to homogeneously deposit a material and minimize the defects in the powder-fed DED. They demonstrated that the predicted models in the methodology should precisely approximate the actual experimental results.(2)The HTCS-150 deposited via the powder-fed DED process showed a hardness as good as HT-H13. Compared to HT-H13, the HTCS-150 also exhibited a relatively high tensile strength at elevated temperatures and a superior wear resistance above 600 °C. It is attributed that the higher molybdenum, vanadium, and tungsten composition of the HTCS-150 leads to mechanical stability in high temperatures.(3)The HTCS-150 deposited via the powder-fed DED process showed a higher thermal conductivity than N-H13 and HT-H13 up to 600 °C, and its thermal conductivity was relative stable up to 800 °C. It is attributed that the large number of interfacial carbides and molybdenum composition in the HTCS-150 leads to a higher thermal conductivity than N-H13 and HT-H13.

The results demonstrate that the powder-fed DED process can additively deposit the HTCS-150 on the typical hot work tool steel with hardened microstructures and enhanced thermal conductivities. However, the difference of the thermal conductivity between the deposited layer and the supporting substrate can cause thermal stress, which can result in different thermal fatigue properties compared to uniformed material. Thus, further studies of the thermal fatigue properties and the fatigue life of the deposited material are desired.

## Figures and Tables

**Figure 1 micromachines-14-00872-f001:**
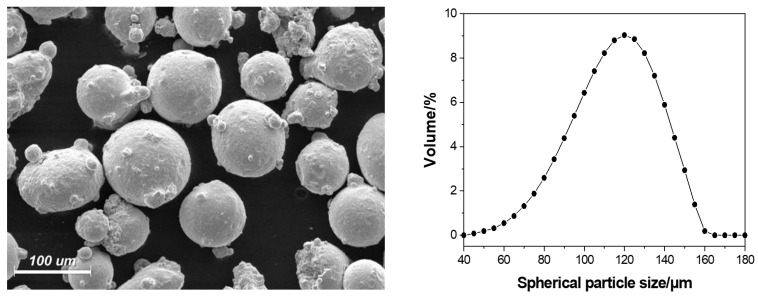
Scanning electron microscope (SEM) image of the HTCS-150 powders and the distribution curve of the particle sizes.

**Figure 2 micromachines-14-00872-f002:**
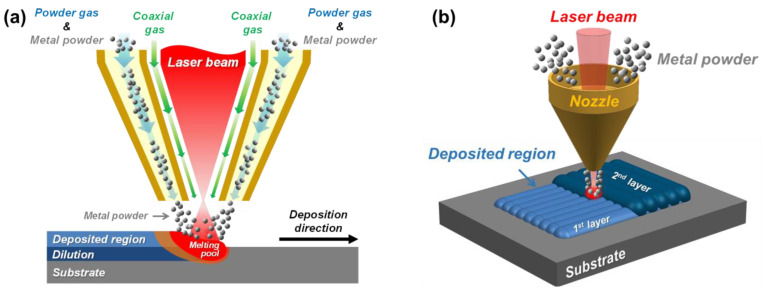
Schematics of (**a**) the powder-fed DED process and (**b**) multi-layered deposition.

**Figure 3 micromachines-14-00872-f003:**
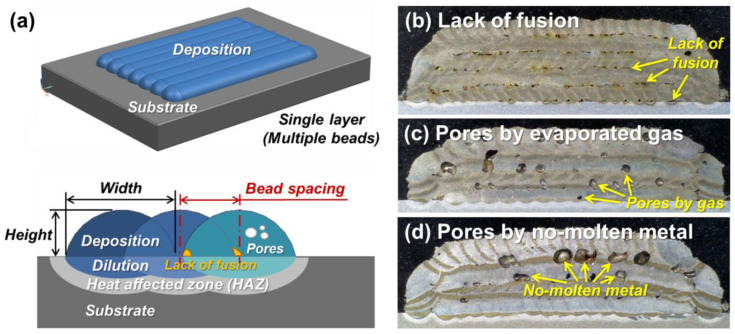
(**a**) Schematics showing the defects (lack of fusion and pores) in the multi-layered depositions. (**b**) A lack of fusion. (**c**) Pores formed by evaporated gas. (**d**) Pores formed by non-molten metal.

**Figure 4 micromachines-14-00872-f004:**
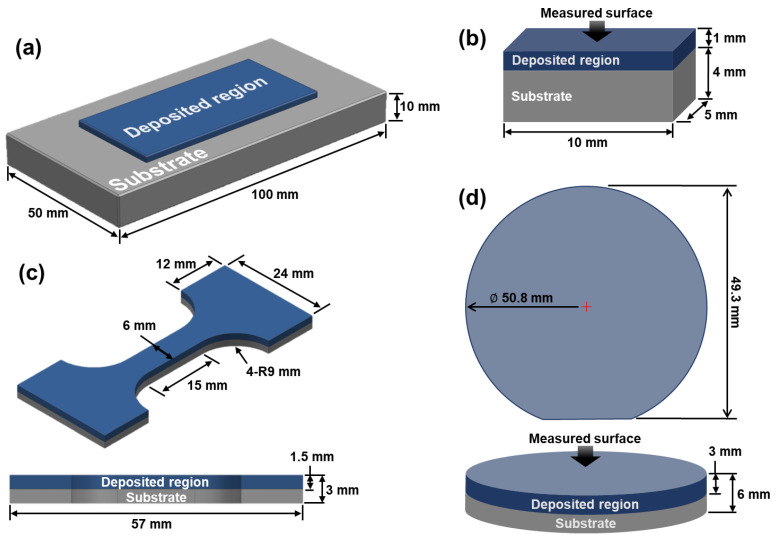
(**a**) Schematic of HTCS-150 locally deposited on H13 substrate. (**b**–**d**) Schematics of test specimens for hardness, tensile, and wear, respectively.

**Figure 5 micromachines-14-00872-f005:**
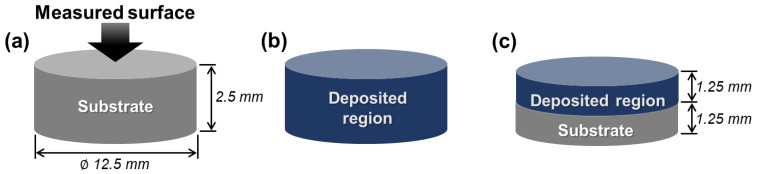
Schematics of test specimens for the measurement of thermal diffusivity. (**a**) H13 substrate. (**b**) As-deposited HTCS150. (**c**) HTCS150 deposited on H13 substrate.

**Figure 6 micromachines-14-00872-f006:**
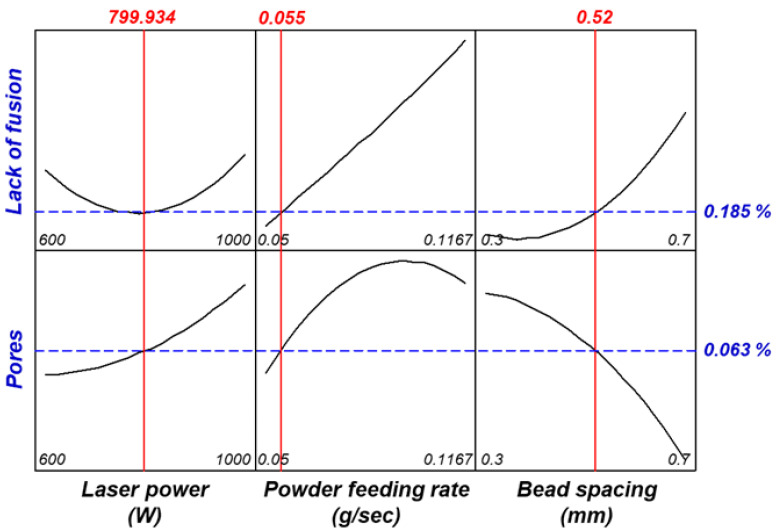
Optimized process parameters of the powder-fed DED for the minimization of the lack of fusion and pores projected onto the response surfaces for the regression models.

**Figure 7 micromachines-14-00872-f007:**
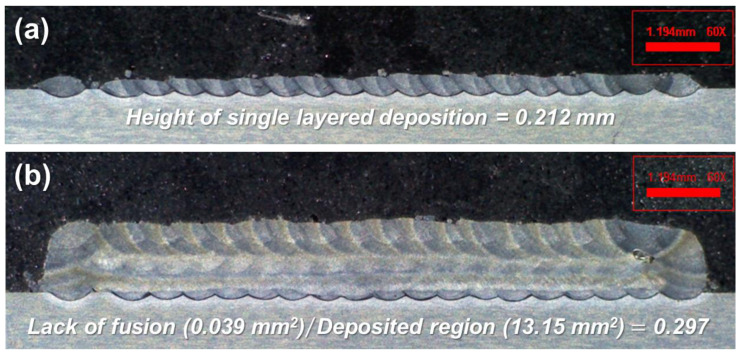
Images of (**a**) single-layered and (**b**) multiple-layered depositions with optimized parameters.

**Figure 8 micromachines-14-00872-f008:**
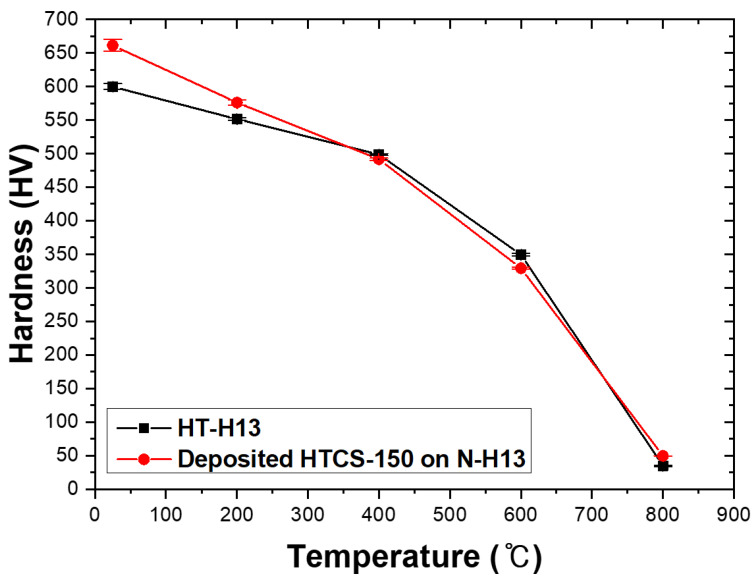
Hardness data of HT-H13 and the HTCS-150 deposited on N-H13 at elevated temperatures.

**Figure 9 micromachines-14-00872-f009:**
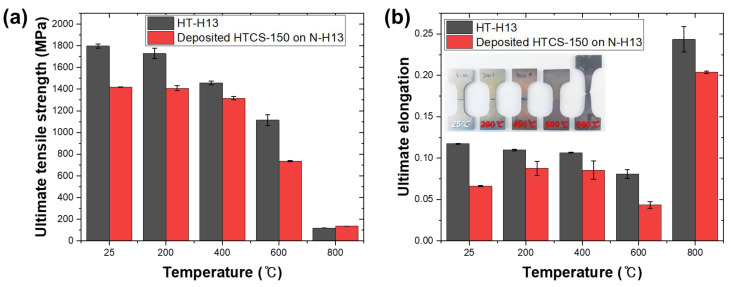
Bar graphs of (**a**) ultimate tensile strength and (**b**) ultimate elongation of the HTCS-150 deposited on N-H13, compared to HT-H13, at elevated temperatures.

**Figure 10 micromachines-14-00872-f010:**
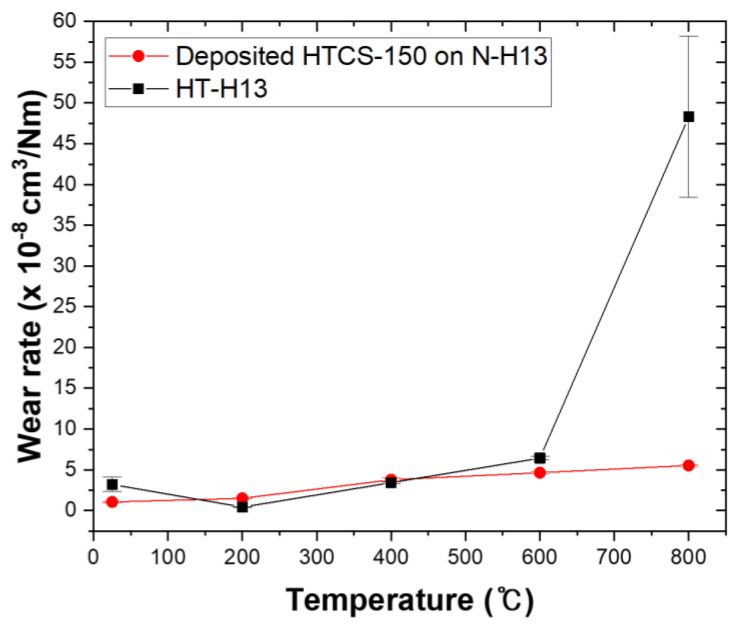
Wear rate of the HT-H13 and the HTCS-150 deposited on N-H13 at elevated temperatures.

**Figure 11 micromachines-14-00872-f011:**
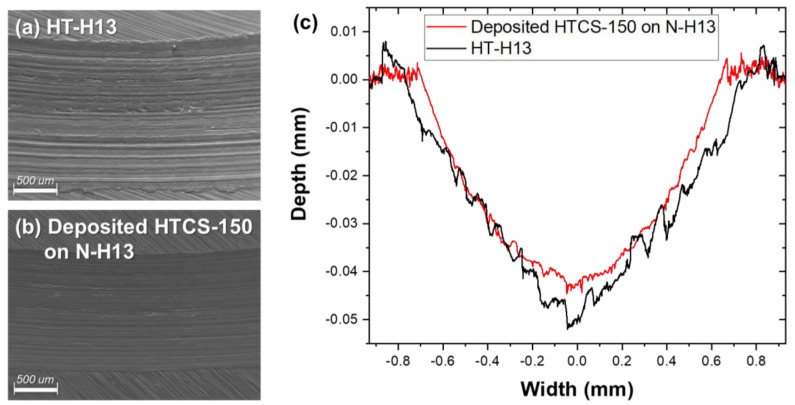
SEM images of the worn surfaces of (**a**) HT-H13 and (**b**) the HTCS-150 deposited on N-H13 at 600 °C. (**c**) Surface profile of the worn surfaces at 600 °C.

**Figure 12 micromachines-14-00872-f012:**
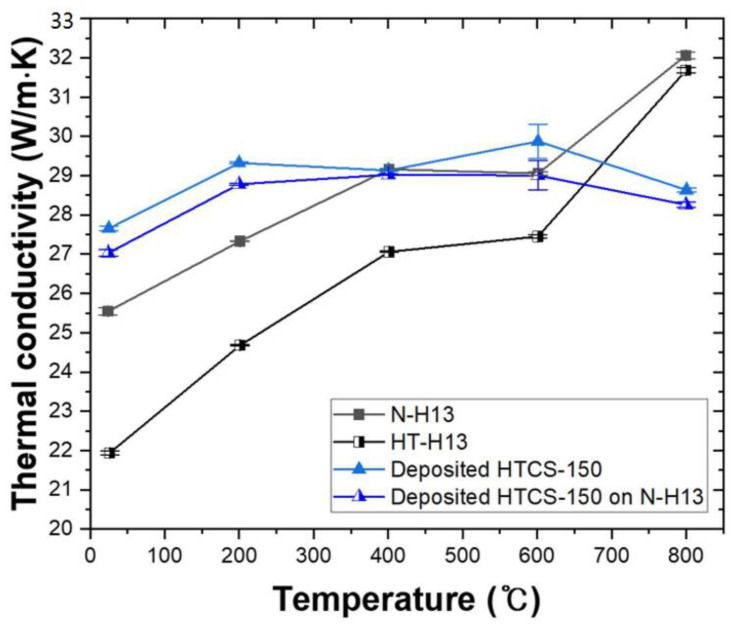
Thermal conductivities of N-H13, HT-H13, as-deposited HTCS-150, and the HTCS-150 deposited on N-13 at elevated temperatures.

**Figure 13 micromachines-14-00872-f013:**
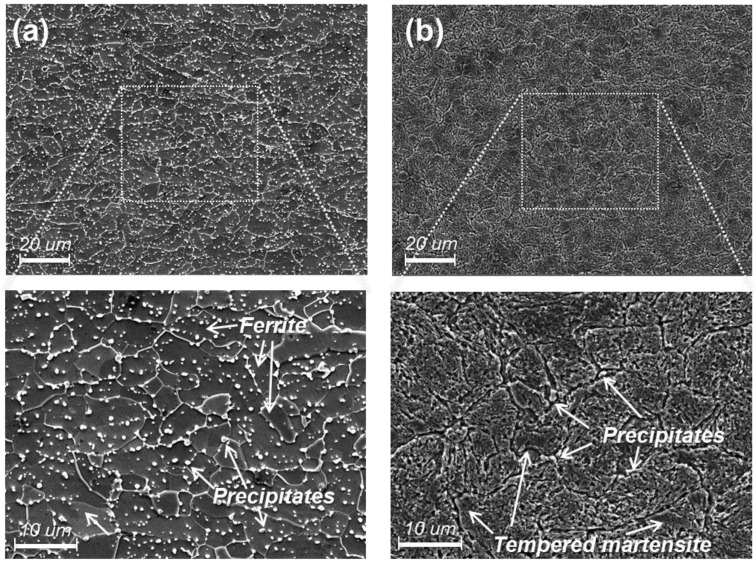
SEM images for the comparison of microstructures of (**a**) N-H13 and (**b**) HT-H13.

**Figure 14 micromachines-14-00872-f014:**
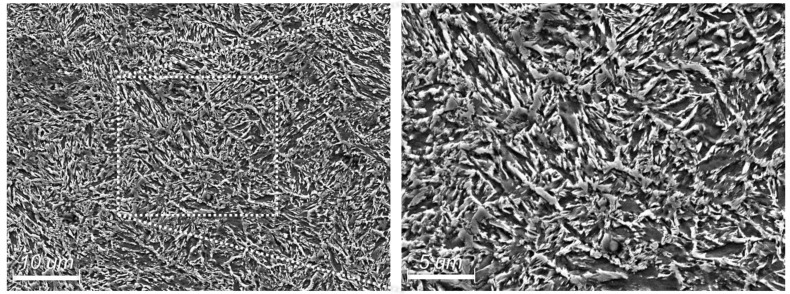
SEM micrographs of the microstructures of the HTCS-150 deposited on N-H13.

**Table 1 micromachines-14-00872-t001:** Chemical compositions of the AISI H13 (N-H13) and HTCS-150 powders [wt.%].

Material	C	Si	Mn	P	S	Mo	Cr	Ni	N	V	Cu	W
AISI H13	0.32	0.8	<0.5	<0.03	0.03	1.00	4.5	<0.25	0.03	0.8	-	-
HTCS-150	0.42	0.12	0.21	0.004	0.016	4.185	0.064	0.029	0.03	-	0.03	3.09

**Table 2 micromachines-14-00872-t002:** Process parameters of the powder-fed DED.

Process Parameters	Value
Laser power	600–1000 W
Powder feeding rate	2–6 g/min
Bead spacing	0.3–0.7 mm
Laser scanning speed (fixed)	800 mm/min
Laser beam diameter (fixed)	1 mm
Power gas (fixed)	2.5 L/min
Coaxial gas (fixed)	8.0 L/min

**Table 3 micromachines-14-00872-t003:** Control variables for experimental cases of CCD and their bounds.

Control Variables	Units	Factor Levels (Point Levels on CCD)				
		(−1.682)	(−1)	(0)	(1)	(1.682)
**Laser power**	W	600	700	800	900	1000
**Powder feed rate**	g/min	3.0	4.0	5.0	6.0	7.0
**Bead spacing**	mm	0.3	0.4	0.5	0.6	0.7

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
