# Peer review of "Mechanical and Thermal Properties of the High Thermal Conductivity Steel (HTCS) Additively Manufactured via Powder-Fed Direct Energy Deposition"

_micromachines, 2023, doi:10.3390/mi14040872_

Round 1

Reviewer 1 Report

This manuscript investigated the mechanical and thermal properties of the high thermal conductivity steel (HTCS) additively manufactured via powder-fed direct energy deposition. I did not find major technical problems. However, some minor problems need to be addressed before the publication of this manuscript.

[1]      The deposited HTCS-150 is comprehensively evaluated through hardness, tensile, and wear tests at the different temperatures of 25, 200, 400, 600, and 800 ℃.Why selected these four temperatures.

[2]      The literature review is roughly (1-4, 5-7, 11-14), and the number is also not enough. More literature research needs to be done.

[3]      ℃change to “°C”, check through the manuscript.

[4]      800℃, a space should added between the number and unit, check through the manuscript.

Author Response

Please, refer to the attachment.

Reviewer 2 Report

The manuscript reports on the production of coated steel with improved thermal conductivity. The analysis of the data is convincing on both sides of material performances and morphological characterization. The manuscript is clear in its presentation and the presented data support the conclusions. Moreover the reading is facilitated by good English style format. For all these reasons the manuscript can be published as is.

Author Response

Please, refer to the attachment.

Reviewer 3 Report

Table 1. Please revise the chemical composition of the HTCS-150 steel, in particular its carbon content (a steel cannot be a steel without carbon).

Lines 87-88. “[…] spherical particles with diameters of 50–100 μm as shown in Figure 1”, but Figure 1 shows a wider range of diameters than 50–100 μm (perhaps 40–160 μm).

Line 132. Eq (2) “Area of lack of fusion […]” instead of “Area of lact of fusion […]”.

Author Response

Please, refer to the attachment.

Round 2

Reviewer 1 Report

Authors have carefully revised the manuscript.